# Developing and Understanding Olfactory Evaluation of Boar Taint

**DOI:** 10.3390/ani10091684

**Published:** 2020-09-17

**Authors:** Evert Heyrman, Steven Janssens, Nadine Buys, Lynn Vanhaecke, Sam Millet, Frank A. M. Tuyttens, Jella Wauters, Marijke Aluwé

**Affiliations:** 1Animal Sciences Unit, ILVO (Flanders Research Institute for Agriculture, Fisheries and Food), Burgemeester Van Gansberghelaan 92, 9090 Melle, Belgium; evert.heyrman@ilvo.vlaanderen.be (E.H.); sam.millet@ilvo.vlaanderen.be (S.M.); frank.tuyttens@ilvo.vlaanderen.be (F.A.M.T.); 2Department of Biosystems, Livestock Genetics, Faculty of Biosceince Engineering, KU Leuven, Kasteelpark Arenberg 20, 3001 Heverlee, Belgium; steven.janssens@kuleuven.be (S.J.); nadine.buys@kuleuven.be (N.B.); 3Ghent University, Faculty of Veterinary Medicine, Department of Veterinary Public Health and Food Safety, Laboratory of Chemical Analysis, B-9820 Merelbeke, Belgium; lynn.vanhaecke@ugent.be (L.V.); jella.wauters@ugent.be (J.W.)

**Keywords:** panel, boar taint, androstenone, skatole, indole, olfactory detection, pigs

## Abstract

**Simple Summary:**

Boar taint is an unpleasant smell and taste of fat of uncastrated male pigs. Growing welfare concerns are pushing towards a ban on the common practice of castrating male piglets as a means to prevent boar taint. This pushes the pork industry to apply alternative strategies to prevent the consumption of tainted of meat. Detecting boar taint is an important aspect of solving this problem, both as a control strategy in slaughterhouses and in boar taint research. This study provides a training protocol and scoring method as well as recommendations for evaluating boar taint.

**Abstract:**

Trained expert panels are used routinely in boar taint research, with varying protocols for training of panelists and scoring methods. We describe a standardized process for training and scoring, to contribute to standardize the olfactory detection of boar taint. Three experiments are described in which we (1) evaluate the importance of training and the effect of the previous sample, (2) determine detection thresholds on strips and in fat for our panel, and (3) test priming panelists before boar taint evaluation. For the final evaluation of boar taint, we propose a consistent three-person evaluation scoring on a 0–4 scale using a final mean score of 0.5 as the cut-off for boar taint. This gave an optimal sensitivity of 0.81 and a specificity of 0.56 compared to chemical cut-offs. Even limited training proved useful, but priming assessors with strips did not improve the evaluation of fat samples. Detection thresholds were higher in fat compared to strips, except for indole. We recommend panelists to always smell a non-tainted control sample after a tainted one as a ‘reset’ mechanism, before continuing. For longitudinal studies, we additionally advise to set up an expert panel with a fixed number of assessors performing each evaluation in duplicate.

## 1. Introduction

Boar taint is an unpleasant odor/taste caused by androstenone (AND), skatole (SKA), and to some extent indole (IND) present in the fat tissue of uncastrated male pigs [1,2,3]. Traditionally male pigs are castrated to prevent this boar taint, but growing welfare concerns are pushing the pig industry towards raising uncastrated or entire male pigs. Preventing tainted meat from reaching the consumer is a primary concern for the pig chain.

Olfactory boar taint evaluation can be used as a boar taint detection system, in experimental settings as well as for an online detection system at a slaughterhouse. Although different training protocols have been described [4], they are all similar in their underlying principles. Anosmia for AND, i.e., the inability to smell AND, is well documented in humans and is a critical criterion for exclusion of a panelist in an olfactory boar taint panel [5]. It has been found that the incidence of anosmia for AND is higher among men than women, and higher among older people. No difference was found between smokers and non-smokers. It is generally advised though to abstain from smoking, as well as eating and drinking half an hour prior to evaluating boar taint [6]. For Belgium 54.7% of subjects have been found to be anosmic for AND [6]. No such anosmia was found for SKA [7]. Testing for AND sensitivity is thus a necessary first step in selecting suitable panelists. This is generally done by offering a discriminatory test with AND on smell strips or dissolved in water. After this selection procedure, retained candidates are acquainted with the odors from heated pork fat and trained with smell strips containing boar taint compounds. They are taught the scoring system and given feedback about their scoring during training until they perform the tests with a minimum of mistakes [8,9,10,11].

When used in experimental settings, olfactory evaluation is mainly performed by a panel of experts to optimize the reliability of the sensory method.

Although olfactory evaluation of boar taint is becoming a routine practice in some (European) slaughterhouses, literature on the repeatability, the detection limits and the effect of training and priming is scarce. It has been found that repeated exposure to AND increases a person’s acuity to that compound. This was tested over a period of six weeks [12]. Intra-rater reproducibility has been evaluated previously and has been found to be ranging from 0.19 to 0.32 which is considered low [9]. Concerning detection limits, for AND and SKA in sunflower oil they have been reported to be 0.21 µg/g and 0.10 µg/g respectively [13]. Variability in sensory thresholds over time has been found in earlier studies on smell strips on subsequent days (ranging seven dilution levels) [14].

In order to better understand and eventually optimize a standardized boar taint scoring method we performed several experiments. First, we evaluated the role of training, combined with the effect of the previous sample on panelist performance by looking at inter and intra rater reliability, sensitivity, and specificity. Second, we determined detection limits of AND, SKA, and IND on smell strips and in the fat of our panel and how these evolved over days. Last, we tested if we could increase panel performance by priming them with AND or SKA smell strips prior to the evaluation and on varying times of the day. Based on the outcomes of these experiments we proposed set of recommendations for a standardized olfactory evaluation of boar taint by a trained panel.

## 2. Materials and Methods

As an overview we describe the 5-step training protocol, the development of the scoring method and finally 3 separate experiments that were set-up to increase insights in the sensory evaluation of boar taint.

### 2.1. Training Protocol

A protocol was developed to train candidates who participated in the olfactory panel. It consists of a 5-step procedure (Figure 1).

#### 2.1.1. Step 1: Selection

Testing the candidate’s sensitivity to both AND and SKA was performed in this step. For each compound 3 series of 3 3-alternate-forced-choice (3-AFC) tests of smell strips were performed, with one strip spiked with a compound and the other two left blank. Each strip was labeled with a random 3-digit code. Each series per compound had a positive sample of 0.5 µg/mL, 5 µg/mL, and 50 µg/mL. The tests were performed in ascending order of concentration. The candidate was asked to identify the positive smell strip.

A candidate passed this selection step when he/she could correctly complete the 3-AFC tests for 50 and 5 µg/mL.

#### 2.1.2. Step 2: Difference and Rank

The candidate was presented with 4 series consisting of 3 smell strips for SKA and AND each. Of these, 2 were spiked with either a low (L: 5 µg/mL) or high (H: 50 µg/mL) concentration, 1 was left blank. Each strip was labeled with a random 3-digit code. The candidate was presented with each strip in random order, but the order was kept constant for all candidates. The candidate was asked to place the smell strips in order of ascending concentration.

In a subsequent test the candidate was presented with 4 series of 5 smell strips each with 2 L and H for AND, 2 L and H for SKA, and 1 left blank. Each strip was labeled with a random 3-digit code. The candidate was presented with each strip in random order, but the order was kept constant for each candidate. The candidate was asked to place the smell strips in order of ascending concentration for each compound separately.

If the candidate could perform this test without fault for 3 consecutive series for both tests, he/she passed this step.

#### 2.1.3. Step 3: Introduction to Fat Samples

In this phase the candidate was first introduced to the scoring system (see below) and the hot iron method (with the soldering iron at 350 °C). The practical aspects were explained such as cleaning the soldering iron between samples and filling in the scoring sheet. Together with the trainer, a mix of 10 fat samples positive for boar taint (T) and 10 negative for boar taint (N) were evaluated (selected based on prior olfactory evaluation). The experience was discussed, and the candidate received feedback on his scoring.

#### 2.1.4. Step 4: 3-AFC Tests of Fat Samples

The first test with fat samples was evaluating 9 series of 3-AFC test with 2 N samples and 1 T sample. Each sample was labeled with a random 3-digit code. The candidate was asked to identify the T sample. In order to pass this step, the candidate had to be able to perform this test with max 2 mistakes and had to be able to correct his/her assessment after receiving feedback on the odd sample and redoing the test in a different order.

#### 2.1.5. Step 5: Series of Fat Samples

In the final test, the candidate was presented with 3 series of 15 fat samples with 5 T and 10 N samples. Each series consisted of replicates of the same samples in different order. The candidate was asked to identify the T samples in each series without knowing how many tainted samples there were. When the candidate was able to perform this test with a max of 2 mistakes and was able to correct his assessment after receiving feedback, he/she was considered passed.

### 2.2. Scoring Method

The scoring method used by our olfactory panel was optimized during the research. This optimization procedure involved 3 improvements. During the first stage we started with a preliminary scale of boar taint intensity and evaluated the method as it was being used (Table 1). This scale was used in a first observational study [15]. During the second stage, we replaced the 8-point scale with a 5-point scale to separate the non-boar taint scale, changed the criteria for the final evaluation of boar taint status, and made practical changes to the scoring method itself based on experience and results from the experiments discussed later. This method was used in the study described in a second observational study [16]. During the final stage used in an experimental study [17], the procedure and scale developed in stage II was applied, only the cutoff used to evaluate boar taint status was adapted to increase sensitivity. The scoring method in the 3 stages is described in the following paragraphs.

#### 2.2.1. Stage I

The scoring system used here was an 8-point scale (1 = no aberrant odor, 2 = some aberrant odor but no boar taint, 3 = aberrant odor but no boar taint, 4 = very light boar taint, 5 = light boar taint, 6 = some boar taint, 7 = strong boar taint, 8 = very strong boar taint). Each fat sample (about 40 g) was evaluated by at least 2 panelists (up to 5 panelists), by each trained panelist independently and blind to each other and to the origin of the neck fat sample, with the median of the panel as final score. A sample was considered tainted (OLF-tainted) if the final score was higher than 3.

To evaluate panelist performance, chemical analysis was done on a representative number of neckfat samples (*n* = 254), from the full scale of olfactory scores. Analysis was done using UHPLC-HR-Orbitrap-MS to determine AND, SKA and IND concentrations (µg/g liquid fat) [18]. Cut-offs were chosen to be 2.0 µg/g, 0.25 µg/g, and 0.15 µg/g respectively [15].

#### 2.2.2. Stage II

All neckfat samples were evaluated by exactly 3 trained panelists (vs. 2–5 in stage I). These panelists participated based on availability from a total of 6. Hence panels had variable composition between but not within slaughter batches. Scoring was done on a 5-point scale (0 = no taint, 1 = light taint, 2 = fair taint, 3 = strong taint, 4 = very strong taint) by each panelist independently and blind to each other and to the origin of the neckfat sample. If the final mean score was equal to or exceeded 1.5, the sample was considered tainted.

A selection of stored samples that received a final median score of 0 (*n* = 97), 1 (*n* = 98), 2 (*n* = 99), 3 and 4 (*n* = 100) were chemically analyzed for AND, SKA, and IND using HPLC-Orbitrap-MS [18].

#### 2.2.3. Stage III

All neckfat samples were evaluated by exactly 3 trained panelists. Using the hot iron method, the panel scored each sample on a 5-point scale (0 = no taint, 1 = light taint, 2 = some taint, 3 = strong taint, 4 = very strong taint). The samples were evaluated twice on 2 following days. If the mean final score was higher than 0.5 the sample was considered tainted. In other words, if half of the 6 scores for a sample received the value of 1, the sample was considered tainted. This was a very stringent selection for detecting even lightly tainted samples.

Chemical analysis was performed on a random selection of samples balanced over the full scale of olfactory scores (*n* = 68) using UHPLC-HR-Orbitrap-MS to establish AND, SKA, and IND concentrations (µg/g liquid fat) [18].

### 2.3. Experiment 1: Effect of Familiarity and Previous Sample

#### 2.3.1. Samples

Neck fat samples (*n* = 30) collected at the slaughterhouse and stored frozen at −20 °C were sliced vertically in thirds, thawed and the same replicate samples were placed in a different order to create 3 series. One series consisted of 8 samples positive for boar taint (T) and 22 negative for boar taint (N). Samples were considered tainted if skatole (SKA) or androstenone (AND) levels (determined by U-HPLC-HR-MS [18]) were above a cut-off of respectively 0.25 and 2.0 µg/g. Chemical analysis revealed a mean SKA and AND level of respectively 0.57 ± 0.43 and 2.4 ± 2.4 µg/g in the tainted samples, and 0.03 ± 0.05 and 0.40 ± 0.40 µg/g in non-tainted samples. Within each series the sample order was such to contain all possible combinations of sample x (evaluated sample) and sample x-1 (previous sample), i.e., N-N, N-T, T-N, T-T.

#### 2.3.2. Scoring and Scale

Participants evaluated all samples using the hot iron method with a 15 min break between series. All participants were instructed to heat each sample with a soldering iron for about 3 s, then smell the sample once and give a score. They were also asked to clean the soldering iron with paper doused with ethanol between each sample. The scoring system consisted of a 5-point scale: 0 (no aberrant smell) to 4 (very strong aberrant smell).

#### 2.3.3. Participants

Participants (*n* = 18) were classified in one of 3 groups; (1) “G1—trained”: 6 trained panelists, (2) “G2—familiar”: 6 untrained people who did have some contact with AND and/or SKA, and/or general boar taint, (3) “G3—unfamiliar”: 6 people had never had any contact with boar taint (compounds). All participants were aware that they were participating in a test concerning boar taint but didn’t know how many samples were tainted. Before starting the test, each participant was given a reference sample without boar taint (barrow) to familiarize them with normal fat odor and to learn using the hot iron method.

### 2.4. Experiment 2: Detection Threshold of AND, SKA, and IND

The general method used is an adaptation of E-679-04 [19] for determining a sensory threshold using multiple concentration steps to finely determine the threshold. Two experiments (one with fat strips and one with spiked fat) were set-up using a series of ascending concentrations of 3-AFC tests (dilution steps) for each compound (AND, SKA, and IND). Each experiment was done using a different matrix: All samples were scored for boar taint by a trained olfactory panel (*n* = 6). None of the participants did more than 2 test series each day and there was a minimum of 2 h between subsequent testing. In total, each participant completed 3 of these tests per matrix and per compound.

#### 2.4.1. On Strips

In this part of the experiment, the compounds were spiked on smell strips. Smell strips (Supplier: Carl Roth, Karlsruhe, Germany, Order no.: 1679.1) and tubes (Supplier: Carl Roth, Karlsruhe, Germany, Order no.: K938.1; Lids, Supplier: Carl Roth, Karlsruhe, Germany, Order no.: E032.1) were labeled with three-digit codes. A 20 µL drop of the appropriate solution was applied to each strip and they were left to dry for 24 h in open tubes under a fume hood. After that the tubes were closed and held at room temperature.

Solutions of AND, SKA, and IND (*n* = 7 each) in propylene glycol were made (µg/mL: 0.05, 0.25, 0.5, 2.5, 5, 25, and 50). The strips with the compounds were offered separately per compound in an ascending order in seven 3-AFC tests. The participants were asked to identify the odd sample for each test as well as indicate the odor intensity on a 10 cm line scale ranging from “no scent” to “very strong scent”.

#### 2.4.2. In Fat

In this part of the experiment, fat from barrows was put in a mixer and spiked with the compounds. Fat from barrows was mixed in a blender and homogenized across samples. Fat samples spiked with AND, SKA, or IND (*n* = 10 for each compound separately, dissolved in propylene glycol) were made (µg/g in fat; AND: 0.8–8.0, in 0.8 steps, SKA and IND: 0.1–1.0, in 0.1 steps). Prior to evaluation, fat samples were kept at 4 °C. The fat samples with the compounds were offered separately per compound in an ascending order in ten 3-AFC tests. The participant was asked to identify the odd sample for each test as well as indicate the odor intensity on a 10 cm line scale ranging from “no scent” to “very strong scent”.

### 2.5. Experiment 3: Priming with Smell Strips and Time of Day

The experiment was organized in a balanced test with or without priming with smell strip effect and the moment of scoring (before or after noon). Participants were trained olfactory panelists (2 women and 3 men). The experiment was repeated twice a day (before and after noon) over two days using 4 replicate series with samples in a different order. There were always 10 samples positive for boar taint (T) and 20 negative for boar taint (N) (as determined by chemical analysis of AND, SKA, and IND). Smell strips (*n* = 2) were prepared, one spiked with AND and one with SKA (both 50 µg/mL). Prior to evaluating the series of fat samples, the panelists were instructed to either first smell both smell strips or to start on the fat samples immediately. Fat samples were stored vacuum packed at −20 °C prior to analysis when they were thawed at 5 °C.

### 2.6. Data Analysis

#### 2.6.1. Scoring Method

A ROC (Receiver operating characteristic) analysis was done for each stage to determine a cutoff score for which sensitivity and specificity were considered desirable [4]. A ROC curve is a tool for diagnostic test evaluation, it plots the sensitivity versus the specificity for different cutoffs for a parameter.

Individual binomial models were used to find the link between olfactory detection by the panelists and the boar taint compound concentrations. This was done for all stages with AND, SKA, and IND as predicting variables and the logit P (boar taint) as predicted value (based on final score). The same analysis was done for all compounds on a smaller sample set where the other compounds were below cutoff values. To account for interactions, three novel variables were set up indicating if a compound was over or under the cutoff. For each combination of these new variables (AND & SKA, SKA & IND and AND & IND) 3 univariate logistic regression models were used to predict the P (boar taint). In this way the “predicting” variable was a four-level factor indicating if both compounds were under cutoff, if one of the two compounds was over cutoff, or if both compounds were over cutoff.

#### 2.6.2. Experiment 1

Sensitivity and specificity were calculated [20] for 3 cutoff scores (CO1: ≥1, CO2: ≥2 or CO3: ≥3) and the chemical thresholds as reference method (≥0.25 SKA and ≥2.0 µg/g AND). A mixed model was used including sample type ((x) and (x − 1)), participant group (G1, 2 and 3), and order (the 1st, 2nd or 3rd series evaluated) as the 3 fixed effects and individual participant and sample as the 2 random effects. All non-significant (*p* ≥ 0.05) effects were sequentially removed by highest *p*-value until only significant effects remained in the model.

#### 2.6.3. Experiment 2

Thresholds were calculated by the method outlined in Lawless and Heymann [21]. First, the geometric mean was calculated of the concentration at the last incorrect test and at the first correct test of an uninterrupted series. This was done for each participant and each test. Then, the geometric mean of the 3 replicate tests for each participant was calculated followed by the geometric mean for all participants per compound.

For both the series of smell strips and of fat, the geometric mean of the last negative and the first positive concentration per panelist was taken as individual threshold. A further geometric mean over all panelists gave the detection threshold per compound.

#### 2.6.4. Experiment 3

The data was analyzed with a linear mixed model, with the scores of the panelists as dependent variables. The time of scoring and the use of smell strips were considered fixed factors, the samples and panelists were considered random effects.

## 3. Results

### 3.1. Scoring Method

#### 3.1.1. ROC Curves to Determine a Cutoff Score

For all three stages, subsamples from the studies were taken to evaluate the olfactory panel performance compared to chemical analysis. The OLF-tainted samples were mainly above cutoff level for SKA (40.7%) and IND (33.6%) in stage I (Table 2). In stage II, samples were high in AND (37.9%) as well as SKA (31.1%) and less for IND (13.7%). For stage II, the highest prevalence of CHEM-positive samples were found, for both AND (14.2) and SKA (11.7%). In stage III, samples were mainly positive for SKA (20.6%) and none of the OLF-non-tainted samples were positive for AND or SKA.

If we use the chemical analysis as the standard method and olfactory evaluation as the predictor, cutoff score for the olfactory assessment were chosen based to optimize sensitivity and specificity (Figure 2). For stage I, sensitivity was 0.76 and specificity was 0.77 at a cutoff level of 3 (scale 1 to 8) based on the median score (Figure 2a). For stage II, sensitivity was 0.72 and specificity was 0.67 at a cutoff level of 1.5 (mean score, given on a scale from 0 to 4) (Figure 2b). In stage III, sensitivity was 0.81 and specificity was 0.56 at a cutoff level of 0.5 (mean score, given on a scale from 0 to 4) (Figure 2c).

#### 3.1.2. Relationship between Olfactory Score and Boar Taint Compounds

Androstenone, skatole, and indole were of significant influence on the odds of a sample being evaluated positively by the olfactory panel of stage I (*p* = 0.044, *p* < 0.001, *p* = 0.001) and stage II (*p* < 0.001, *p* < 0.001, *p* < 0.001) (Figure 3). Which is to be expected as the olfactory cut-off is chosen on ROC comparison with these compounds. In stage III, AND (*p* = 0.204), and IND (*p* = 0.858) had no effect, while SKA (*p* = 0.002) did have a significant effect on the odds of a sample being evaluated positively by the olfactory panel. Interaction terms between boar taint compounds were not significant for stage I and stage II (Figure 4). For stage III, the number of samples was too low to test the interaction model.

For AND, the chance of an OLF-tainted carcass showed a linear increase with increasing AND level for all three stages (Figure 3). For stage I, AND concentration at which the chance for an OLF-tainted carcass was 50%, was at 2.0 µg/g AND. For stage III, this was at an AND concentration of 1.5 µg/g. For SKA, the chance to have an OLF-tainted sample varied from 0.2 at the lowest SKA content up to 1.0 at the upper content with a plateau at 0.5 µg/g SKA (stage I, stage II, and stage III). At a SKA concentration of 0.25 µg/g, the chance for an OLF-tainted carcass was 50% (stage II). The odds of an OLF-tainted sample increased with increasing IND concentration, with a plateau reached at higher concentration (0.4 µg/g) in stage I and stage II. For stage III, effect of IND was linear. At an IND content of 0.15 µg/g, the chance for an OLF-tainted sample is around 50% for these stages. (Figure 2 and Figure 3).

### 3.2. Experiment 1: Familiarity and Effect of Preceding Sample

The average score given by participants for all sample types increases progressively for participant groups with decreasing familiarity with boar taint, with groups with lesser familiarity (G2–G3) giving progressively higher scores (*p* = 0.002). For all groups of participants, the average score was lower if the preceding sample was tainted (*p* < 0.001, Figure 5).

The interaction effect between type of participant group and the order of the presented series was significant (*p* = 0.005), reflecting that G3 and G2 gave a higher average score to samples from the first and second series compared to the following series they received (Table 3). Inter (consistency between raters) and intra (consistency within raters) rater reliability increased with increasing training and familiarity from 0.16 to 0.45 and from 0.18 to 0.53, respectively (Table 3). With lowering familiarity, sensitivity increased, and specificity decreased. Sensitivity and specificity also depended on the cutoff score chosen (CO1, CO2, or CO3). With increasing cutoff score, specificity increased mainly for G2 and G3, while sensitivity decreased in all three groups.

### 3.3. Experiment 2: Detection Threshold AND, SKA, and IND

The estimated threshold for the respective compound is shown in Table 4. Thresholds per panelist and per day were plotted to illustrate the variation between and within panelists on smell strips (Figure 6) and in fat samples (Figure 7). For all compounds, the thresholds varied considerably within and between panelists.

### 3.4. Experiment 3: Priming with Smell Strips

There was no significant effect of before- or after-noon (*p* = 0.123), or with or without priming with smell strips (*p* = 0.735) on the average score given for boar taint positive fat samples. The average scores without strips were 1.18 before and 0.97 after noon. The average scores with strips were 1.13 before and 0.93 after noon.

There was also no significant effect of before- or after-noon (*p* = 0.700), or with or without priming with smell strips (*p* = 0.248) on average score given for boar taint negative fat samples. The average scores without strips were 0.025 before and 0.015 after noon. The average scores with strips were 0.055 before and 0.045 after noon.

## 4. Discussion

Training was based on smell strips with SKA and AND, and fat samples with known concentrations of these compounds. The role of IND is still under debate and therefore not specifically included in the training protocol, but was however considered when evaluating the panel performances compared to chemical analysis and threshold levels (experiment 2) were also determined for IND.

As a consequence of the evolution of the scoring system, comparing the prevalence between the different studies is not straightforward. However, changes in the scoring seemed necessary for several reasons. There are 6 main points where the scoring method stages differ. First, exactly 3 trained panelists per sample were used in stage II and III instead of 2–5 panelist in stage I, since the number of panelists was found to be positively correlated to the boar taint risk. Second, mean score was used as final score in stage II and III instead of the median score in stage I as this gave better correspondence with the chemical analysis. Third, a five-point scale was used in stage II and III instead of an 8-point scale in stage I. This was mainly done based on the panelists feedback to make the scale easier to use. Also, distinction between boar taint and other non-boar taint aberrant odors was made more clearly compared to the previous scoring system. This previous ambiguity was confusing to some panelists which led to inconsistencies between them. Fourth, the final cutoff score changed from ≥3 in stage I, ≥1.5 in stage II, and ≥0.5 in stage III. This was mainly a consequence of progressively favoring sensitivity over specificity (see below). Fifth, the total number of individual panelists (seven during stage II instead of four during stage I) changed as well. Sixth, the farms evaluated in the different studies are not all the same, and more slaughter batches per farm were evaluated in a wider time span during stage II compared to stage I, while stage III scoring was used in an experimental study. Also farm related factors may have changed by the time stage II scoring was fully implemented. Finally, the range of boar taint compounds in the sub-samples was different.

With a sensitivity of 0.76, 0.72, 0.81, and a specificity of 0.84, 0.67, and 0.56 for stages I, II, and III respectively, we find the general performance of the olfactory panel sufficient since the parameter are comparable with values for sensitivity (0.16 to 0.61) and specificity (0.97 to 0.82) for trained panels used in other studies [9,22]. Overall when considering the evolution of the scoring system it favors sensitivity over specificity. This results in generally higher boar taint prevalence and thus higher statistical power of the analyses, which was partly a conscious goal, especially during stage III. By this stricter evaluation, even lightly tainted samples are picked up, and the trade-off is that relatively more samples are likely false positives, which might not be desirable for slaughterhouses who on the other hand should avoid false negatives.

When SKA was on a high level (above 0.4 μg/g for all stages), almost all panelists evaluated the sample as tainted but when AND was on a high level in stage I (above 4 μg/g) only 60% of the panelists evaluated it as tainted. For stage II and III 100% of the panelists considered a sample as tainted when AND was at 2.5 μg/g. This could mean that SKA better predicts boar taint for these samples. Differences in ability to detect AND and SKA have been shown for trained panels from different countries using different training protocols [23]. This stresses the importance of selecting for high AND sensitivity when constructing a trained panel [24]. There was a numerical but non-significant difference between the odds of having an OLF-tainted sample when AND alone was high and SKA alone was high (for both stage I and II). Skatole has been indicated in earlier studies as playing a larger role [9,22]. The lower volatility of AND might mean it is perceived later while all samples were evaluated directly after heating [25]. This also stresses the importance of the timing of smelling when heating a sample and taste tests. Between compounds, there were no significant interactions, and this suggests an absence of an additive or inhibiting effect between boar taint compounds in neither stage I and II. This contrasts to some extent with earlier studies where SKA and IND contribute to the perceived intensity of AND, but this effect was not found between SKA and IND was found [26,27]. This could be because only a few (three each) samples in our study had a combination of boar taint compounds (AND & SKA, and AND & IND) above cutoff. It has been posed that the qualitative differences in odor of mixing odorants related to boar taint in varying concentrations can lead to errors by a panel [28]. Given the low prevalence of boar taint, finding enough of these differing fat samples complicates the training and re-training of a long-term panel. Furthermore, differences in the fat matrix itself of the samples in relation to water and fat content and fatty acid composition [29] can influence the release of volatile compounds and thereby the sensory profile [30,31].

As expected, trained panelists had a higher reliability than the other participant groups. Repeated exposure to AND has been shown to lead to higher sensitivity to this compound [12], which could in part explain this. Even the familiar participant group showed a higher reliability than the unfamiliar group indicating that even limited exposure to boar taint (compounds) is enough to identify it more confidently. It has been shown previously for a consumer panel that an initial exposure to boar taint makes assessors more sensitive in evaluation [32]. The same goes for other odorants, when subjects are familiar with an odor, these subjects can perceive this odor with a higher intensity [33]. In consumer studies generally low sensitivity to boar taint has been reported [34]. Results for the sensitivity and specificity are comparable for all three participant groups depending on what cutoff is chosen (Table 3). The generally higher scores for less trained groups both for T and N samples, could indicate that less trained or familiar participants have trouble using the scoring scale and tend to give scores (>0) to normal pig fat which trained panelist simply consider as background smell (score 0).

A sample evaluated after a T sample was scored significantly lower by all participant groups. This could indicate a saturation of the olfactory capacity. After this experiment, our panelists were instructed to always smell a confirmed N sample after a T sample before continuing during future evaluations.

It was hypothesized that priming panelists with smell strips spiked with AND and SKA may improve panel performance if it would enhance the identification of AND and SKA and further increase the aspect of “familiarity”. Contrary it may reduce the ability of panelist to perceive AND and SKA as shown in the same experiment, as panelists are less able to identify a tainted sample after rating another tainted sample. Priming with smell strips was neither disadvantageous nor advantageous. The numerically higher scores in experiment 3 for samples smelled before noon are concordant with the generally accepted optimal time for sensory testing, which is before noon [21].

The detection thresholds varied between panelists and within panelists when testing on different days. Inter- and intra-rater variability over time has been found in earlier studies [14]. Further experiments with more assessors but the same concentrations should enable determining a threshold above which a certain percentage of assessors will consider a carcass tainted. Panelists are most sensitive to skatole in both tests as is concordant with the findings in the previous section. For smell strips, panelists are least sensitive to indole reflecting less familiarity with this compound. For spiked pig fat, thresholds for skatole and indole are within expectation according to the range generally in literature while the threshold for androstenone was much higher. This could indicate that this test is less representative for the release of androstenone from real pig fat when heated. Also, AND was related to a positive evaluation of the sensory panel in the previous section, suggestion that detection of AND is less straightforward compared to SKA.

Thresholds vary between and within panelists over the replicate tests. This is due to inter- and intra-rater variability. A previous study looking at detection thresholds of AND and SKA in spiked sunflower oil found thresholds of 0.05 to 0.2 µg/g for SKA and 0.1 to 0.5 µg/g for AND [13]. This is in accordance with the thresholds found here, at least for strips. Skatole has been found to be detectable at as low as 0.1 µg/g and androstenone at 0.2 µg/g in fat [13,35,36].

## 5. Conclusions

Based on our findings for sensitivity and specificity, our optimized training protocol proved to be suitable for the training of an expert panel. Generally, a couple of recommendations can be made for further standardization, e.g., the number and identity of panelists is best kept constant as sensory evaluation of samples showed to be susceptible to the number of panelist and individual acuity of panelists. Defining a cutoff score showed to be challenging but was facilitated by performing a ROC-analysis. In our studies, SKA was picked up with more consistency compared to AND and was therefore a more reliable and reproducible boar taint predictor in our training model. At the used chemical cutoffs of 2.0 µg/g for AND, 0.25 µg/g for SKA, and 0.15 µg/g for IND, the chance for olfactory detection by the panel was 50%. Our method for determining the threshold for detectable boar taint in fat samples succeeded as a proof of concept but needs adaptation for definition of the AND threshold, since adaptive mechanisms on the individual level in the success of detecting this odour and its intensity seems to be in play. Nevertheless, smelling strips before evaluating samples did not seem to improve panel performance.

## Figures and Tables

**Figure 1 animals-10-01684-f001:**
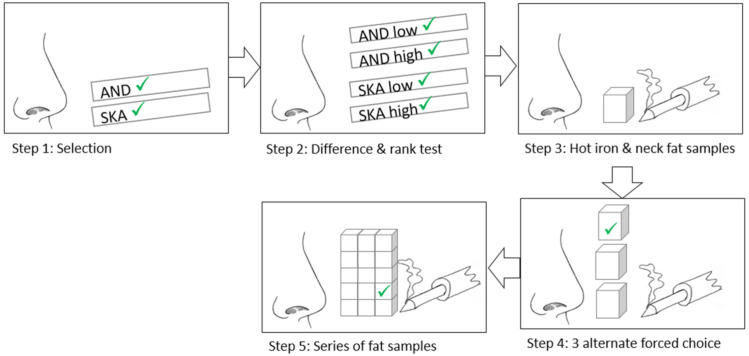
Illustration of the selection and training protocol for panelists of an olfactory panel for boar taint. In the first step there is a selection for SKA and AND sensitivity using smell strips. In the second step panelists are asked to discriminate between AND and SKA and rank smell strips according to concentration. In the third step the hot iron method and scoring system on neck fat samples is taught. In the fourth, 3 alternate forced choice tests are performed where panelists must identify a positive sample among two negative samples. In the fifth step the panelists are asked to smell 3 series of fat samples and identify positive samples.

**Figure 2 animals-10-01684-f002:**
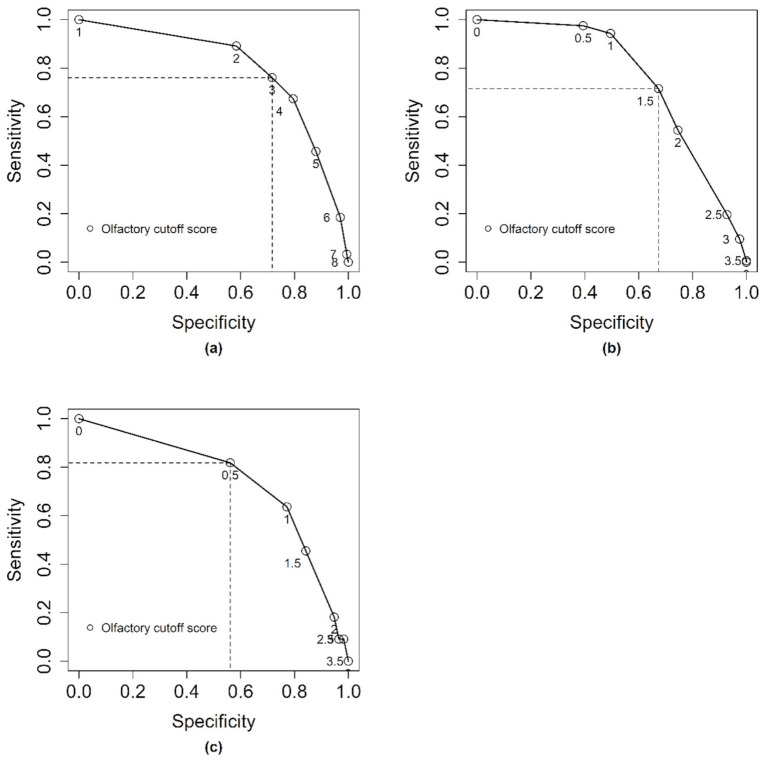
ROC analysis for a selection of neckfat samples from entire male pigs from stage I (**a**), stage II (**b**), and stage III (**c**) of olfactory detection method optimization. Sensitivity and specificity are shown for each possible olfactory cutoff score compared to exceeding the cutoff levels for androstenone (cutoff = 2.0 µg/g), skatole (cutoff = 0.25 µg/g), and indole (cutoff = 0.15 µg/g).

**Figure 3 animals-10-01684-f003:**
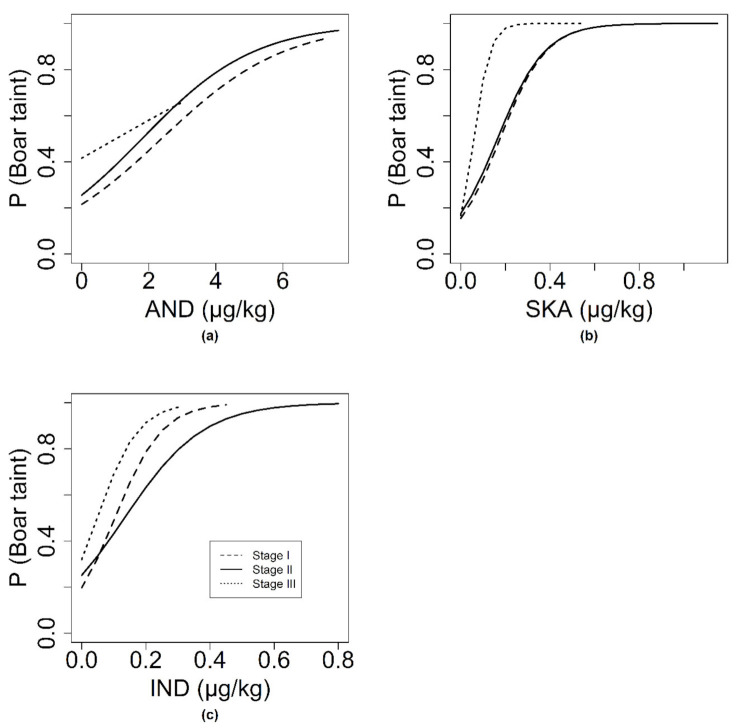
For a subset of entire male pigs, the link between boar taint compounds and the olfactory evaluation for the three stages of olfactory detection method optimization: (**a**)For samples with SKA an IND under cutoff, regression of P (Boar taint) as a function of AND. (**b**) For samples with AND an IND under cutoff, regression of P (Boar taint) as a function of SKA. (**c**)For samples with AND an SKA under cutoff, regression lines of P (Boar taint) as a function of IND. P (Boar taint) indicated the odds that tainted sample be considered tainted by the olfactory panel.

**Figure 4 animals-10-01684-f004:**
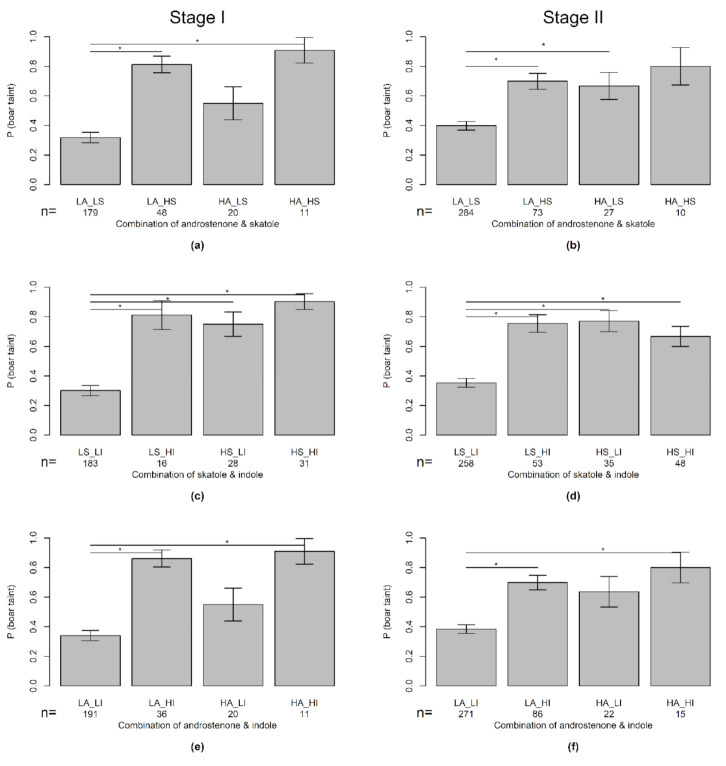
Mean odds of a sample being considered tainted (P (boar taint) by the olfactory panel for combinations of boar taint compounds with concentration lower than (L) of higher than (H) cutoff level of 2.0 µg/g, for androstenone (A), 0.25 µg/g for skatole (S) and 0.15 µg/g indole (I); (**a**) + (**b**) combination of androstenone and skatole stage I + stage II, (**c**) + (**d**) skatole and indole, stage I + stage II, (**e**) + (**f**) androstenone and indole, stage I + stage II. The lines represent significant differences (* *p* < 0.05)—interaction between androstenone, skatole, indole was not significant.

**Figure 5 animals-10-01684-f005:**
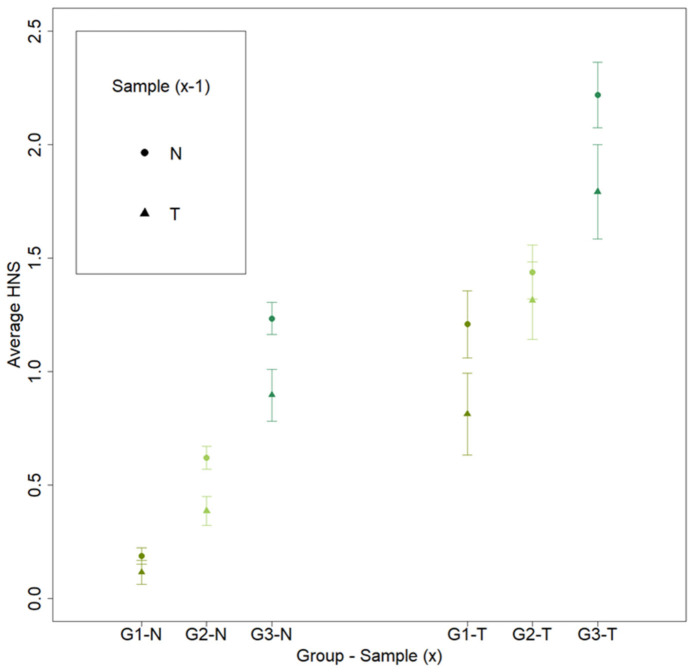
Average score (HNS) for each sample type (x) and previous sample type (x-1) per participant group, consisting of (1) “G1—trained” (6 trained panelists), (2) “G2—familiar” (6 people had some notion of AND and/or SKA, and/or boar tainted samples in general but who were not trained), and (3) “G3—unfamiliar” (6 people who had no notion of boar taint). The average score given by participants for all sample types increases progressively for participant groups with decreasing familiarity with boar taint. For all groups of participants, average score was lower if the preceding sample was tainted.

**Figure 6 animals-10-01684-f006:**
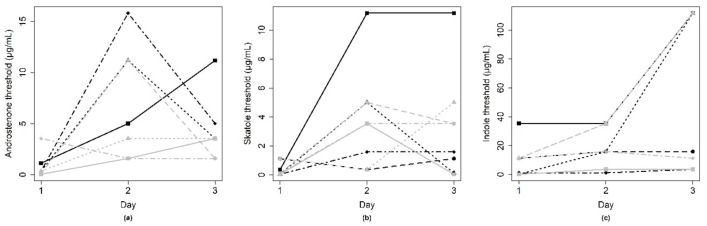
Sensory threshold on smell strips, with each line representing the individual thresholds (*y*-axis) of a panelist for replicate tests on three different days (*x*-axis). For all compounds, (**a**) androstenone, (**b**) skatole, (**c**) indole, the thresholds varied considerably within and between panelists.

**Figure 7 animals-10-01684-f007:**
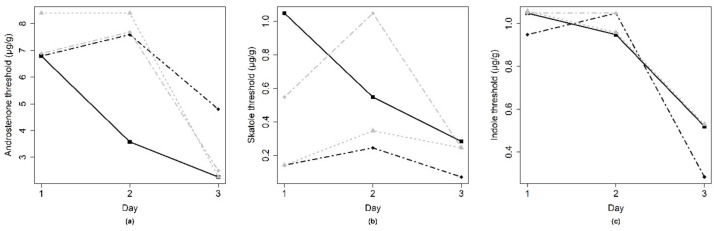
Sensory threshold in fat samples, each line represents the individual thresholds (*y*-axis) of a panelist for replicate tests on three different days (*x*-axis). For all compounds, (**a**) androstenone, (**b**) skatole, (**c**) indole, the thresholds varied considerably within and between panelists.

**Table 1 animals-10-01684-t001:** Comparison of the scoring system used in the study by the olfactory panel (with hot iron method) during the study in stage I, II, and III.

	Stage I	Stage II	Stage III
Scale	8-point	5-point	5-point
	1: no aberrant odor	0: no taint	0: no taint
	8: very strong boar taint	4: very strong taint	4: very strong taint
Separate non-boar taint scale	No	Yes	Yes
Number of panelists/sample	2–5	3	3
Number of panelists available	7	4	4
Repeated scoring (2 scores/panelist/sample)	No	No	Yes
Final score	Median	Mean	Mean
Cutoff score	3	1.5	0.5
Sensitivity	0.76	0.72	0.81
Specificity	0.77	0.67	0.56

**Table 2 animals-10-01684-t002:** Summary of subsample of chemically and olfactory evaluated samples from three stages of improvement of the scoring system. Chemical cutoffs were 2.0 µg/g for AND, 0.25 µg/g for SKA, and 0.15 µg/g for IND (CHE tainted).

	Stage I	Stage II	Stage III
For all samples *(n)*	254	394	68
% OLF-tainted	44.5	48.5	50
% CHE-tainted	34.6	40.1	22.1
For OLF-tainted samples (*n*)	113	190	34
% >AND	15.8	37.9	5.9
% >SKA	40.7	31.1	20.6
% >IND	33.6	13.7	5.9
% >ALL	5.3	4.2	0.0
For OLF-non-tainted samples (*n*)	141	204	34
% >AND	7.1	14.2	0.0
% >SKA	7.1	11.7	0.0
% >IND	4.3	3.4	17.6
% >ALL	0.7	1.0	0.0

**Table 3 animals-10-01684-t003:** Inter and intra rater reliability, and sensitivity and specificity for each participant group (G1: trained, G2: familiar, G3: unfamiliar) for cutoff score 1, 2 and 3.

	G1	G2	G3	All
Inter rater reliability	0.45	0.29	0.16	0.25
Intra rater reliability	0.53	0.42	0.18	0.30
Cut-off score 1				
Sensitivity	0.45	0.72	0.82	0.66
Specificity	0.91	0.59	0.40	0.63
Cut-off score 2				
Sensitivity	0.34	0.44	0.62	0.46
Specificity	0.94	0.90	0.66	0.83
Cut-off score 3				
Sensitivity	0.21	0.21	0.40	0.27
Specificity	0.98	0.95	0.84	0.92

**Table 4 animals-10-01684-t004:** Olfactory detection thresholds for AND, SKA, and IND. Determined for 6 trained panelists using smell strips and spiked pig fat.

Compound	Threshold Smell Strips (µg/mL)	Threshold Pig Fat (µg/g)
AND	0.24	6.92
SKA	0.18	0.35
IND	3.71	0.90

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
