# Peer review of "Developing and Understanding Olfactory Evaluation of Boar Taint"

_animals, 2020, doi:10.3390/ani10091684_

Round 1

Reviewer 1 Report

The manuscript presents interesting results on the sensory evaluation of the presence of boar taint in pork. This aspect is of vital importance in the detection of odor when uncastrated or immunoccastrated males are processed in slaughterhouses. Although the recommendations are presented to an expert panel, several of them can be considered for use in slaughterhouses. However, the experimental design and the presentation of the results is confusing, it is not well justified why IND detection is important and why it is used only in some experiments. The methodology is not very detailed in the detection procedure with the panelists, how were the fat samples handled, how were they burned with the hot iron at what T°, were they fortified with the compounds or mixtures of them, how were the compounds combined in the fat and in the strips, why were so many concentrations used, leaving unclear aspects for the evaluation. Many editing errors.

Author Response

Reviewer #1:

The manuscript presents interesting results on the sensory evaluation of the presence of boar taint in pork. This aspect is of vital importance in the detection of odor when uncastrated or immunoccastrated males are processed in slaughterhouses. Although the recommendations are presented to an expert panel, several of them can be considered for use in slaughterhouses. However, the experimental design and the presentation of the results is confusing, it is not well justified why IND detection is important and why it is used only in some experiments.

This is a valid point and we added this justification in the discussion section on line 413: “For the training protocol we only used AND and SKA. The role of IND is not entirely clear, but was taken into account but to further take it into account we did include it in the evaluation of our panel and in the second experiment concerning the sensory thresholds. In the first and third experiment we did not include IND partly to keep it simple for participants.”

The methodology is not very detailed in the detection procedure with the panelists, how were the fat samples handled, how were they burned with the hot iron at what T°, were they fortified with the compounds or mixtures of them, how were the compounds combined in the fat and in the strips, why were so many concentrations used, leaving unclear aspects for the evaluation. Many editing errors.

We included the handling of the fat samples on line 237: “Prior to evaluation fat samples were kept at 4 °C.”

We included the temperature of the soldering iron on line 118: “In this phase the candidate was first introduced to the scoring system (see below) and the hot iron method (with the soldering iron at 350 °C).”

We included that it concerns single compounds on line 236 :” Fat samples spiked with AND, SKA, and or IND (n= 10 for each compound separately, dissolved in propylene glycol) were made (µg/g in fat; AND: 0.8 – 8.0, in 0.8 steps, SKA and IND: 0.1 – 1.0, in 0.1 steps).”

We indicated why multiple steps were used on line 215 : “The general method used is an adaptation of E-679-04 [18] for determining a sensory threshold using multiple concentration steps to finely determine the threshold.”

Reviewer 2 Report

The paper is a sound paper. The importance of a standardized method to detect boar taint is evident. Especially, the ROC has value for the slaugtherhouses.

The described method is well explained and systematic and well developped. The method is suitable for scientific research projects. The paper would have gained in impact if comparisons would have been made with current methods in slaugtherhouses. 

Scientically this is a sound paper.

Minor comments:

line 44 ref. not mentioned

line 92 wereperformed

line 437 and 441 confusing

Author Response

Reviewer #2:

The paper is a sound paper. The importance of a standardized method to detect boar taint is evident. Especially, the ROC has value for the slaugtherhouses.

The described method is well explained and systematic and well developped. The method is suitable for scientific research projects. The paper would have gained in impact if comparisons would have been made with current methods in slaugtherhouses.

Scientically this is a sound paper.

Minor comments:

line 44 ref. not mentioned

Thank you for pointing this out, we added the reference on line 44

line 92 wereperformed

We corrected this error on line 96

line 437 and 441 confusing

We altered the wording and corrected some errors to make these sentences more readable in line 468-477: “It has been shown previously for a consumer panel that an initial exposure to boar taint makes assessors more sensitive in evaluation [32]. The same goes for other odorants that, when subjects are familiar with an odor, that these subjects can perceive this odor with a higher intensity [33]. In consumer studies generally low sensitivity to boar taint has been reported [34]. Results on for the sensitivity and specificity are comparable for all 3 participant groups depending on what cutoff is chosen (Table 3). ). In consumer studies generally low sensitivity to boar taint has been reported [34].The generally higher scores for less trained groups both for T and N samples, could indicate that less trained or familiar participants have trouble using the scoring scale and tend to give scores (>0) to normal pig fat which trained panelist simply consider as background smell (score 0).”

Reviewer 3 Report

A majority of abattoirs which slaughter male carcases identify the level of boar taint in the fat tissue using humans' sense of smell (heat the pig backfat and smell the odour with very well-trained people). In the majority of cases, the tainted male carcases are cheaper in the European market and the boar tainted meat and fat are oriented towards the low-cost pork processing industry. Reducing boar tainted percentage is an important economic challenge for the pig food chain in Europe and in the world pig market.

The manuscript "Developing and understanding olfactory evaluation of boar taint” fit the scope of “Animals” Journal because of providing the relevant data concerning traits very important for slaughterhouses and meat consumers. Presented results are complementing the pool of knowledge concerning the olfactory evaluation of boar taint. However it is necessary to find alternatives to removing the boar taint and preventing the devaluation of pork meat.

Title of the manuscript is adequate to the its text. The objective of the study was clearly described and documented. The investigations were done on sufficient animal material using widely accepted methods. The obtained results are adequately, detailed and sufficient discussed. Conclusions are basing on the presented results.

However, I have some remarks on the manuscript which I would like to be considered before publication.

line 44: what is “...(ref)...”

lines 147-148  you wrote that the experiment was conducted by 2 panelists (up to 5 panelists) and (Table 1) whereas in the line 157 was “3 trained panelists”, whereas in the lines 391, 392 was panelists “(6 during stage II instead of 8 during stage I)…”. What is the correct number ? Please check and explain the discrepancy.

line 240 The acronym ROC is presented for the first time, without presentation of its meaning!

line 323 is “P(boar taint) is…”, correct “P(Boar taint) is…”.

line 441 is “(score  0”,  correct “(score  0)”.

lines 567 and 570  is “Annor-Frempong, I..;”, correct “Annor-Frempong, I.E.;”.

Author Response

Reviewer #3:

A majority of abattoirs which slaughter male carcases identify the level of boar taint in the fat tissue using humans' sense of smell (heat the pig backfat and smell the odour with very well-trained people). In the majority of cases, the tainted male carcases are cheaper in the European market and the boar tainted meat and fat are oriented towards the low-cost pork processing industry. Reducing boar tainted percentage is an important economic challenge for the pig food chain in Europe and in the world pig market. The manuscript "Developing and understanding olfactory evaluation of boar taint” fit the scope of “Animals” Journal because of providing the relevant data concerning traits very important for slaughterhouses and meat consumers. Presented results are complementing the pool of knowledge concerning the olfactory evaluation of boar taint. However it is necessary to find alternatives to removing the boar taint and preventing the devaluation of pork meat. Title of the manuscript is adequate to the its text. The objective of the study was clearly described and documented. The investigations were done on sufficient animal material using widely accepted methods. The obtained results are adequately, detailed and sufficient discussed. Conclusions are basing on the presented results. However, I have some remarks on the manuscript which I would like to be considered before publication.

line 44: what is “...(ref)...”

Thank you for pointing this out, we added the reference on line 44

lines 147-148 you wrote that the experiment was conducted by 2 panelists (up to 5 panelists) and (Table 1) whereas in the line 157 was “3 trained panelists”, whereas in the lines 391, 392 was panelists “(6 during stage II instead of 8 during stage I)…”. What is the correct number ? Please check and explain the discrepancy.

The 2-5 panelist is for stage I while 3 panelist is for stage II. We added this comparison again in the text on line 165 to make this clearer: “All neckfat samples were evaluated by exactly 3 trained panelists (vs. 2-5 in stage I).”

We corrected the number of available panelist on line 420 to correspond to Table 1: “Fifth, the total number of individual panelists (7 during stage II instead of 4 during stage I) changed as well.”

line 240 The acronym ROC is presented for the first time, without presentation of its meaning!

We added this on line 254: “A ROC (Receiver operating characteristic) analysis was done for each stage to determine a cutoff score for which sensitivity and specificity were considered desirable [4].”

line 323 is “P(boar taint) is…”, correct “P(Boar taint) is…”.

We corrected this typo on line 348: “P(boar Boar taint) is the chanceindicated the odds that the olfactory panel would consider an entire male carcass to be taintedtainted sample be considered tainted by the olfactory panel.”

line 441 is “(score 0”, correct “(score 0)”.

We corrected this typo on line 477: “The generally higher scores for less trained groups both for T and N samples, could indicate that less trained or familiar participants have trouble using the scoring scale and tend to give scores (>0) to normal pig fat which trained panelist simply consider as background smell (score 0).”

lines 567 and 570 is “Annor-Frempong, I..;”, correct “Annor-Frempong, I.E.;”.

We corrected this on line 603 and 606

Round 2

Reviewer 1 Report

I have no more suggestions